# Effects of Aquatic Training in Children with Autism Spectrum Disorder

**DOI:** 10.3390/biology11050657

**Published:** 2022-04-25

**Authors:** Hamza Marzouki, Badis Soussi, Okba Selmi, Yamina Hajji, Santo Marsigliante, Ezdine Bouhlel, Antonella Muscella, Katja Weiss, Beat Knechtle

**Affiliations:** 1High Institute of Sports and Physical Education of Kef, University of Jendouba, Kef 7100, Tunisia; hamzic_30@hotmail.com (H.M.); badis.delrio@gmail.com (B.S.); okbaselmii@yahoo.fr (O.S.); 2Halim Professional Training Center for Young People with Autism, Ariana 2091, Tunisia; yamina_haji19@hotmail.com; 3Department of Biological and Environmental Science and Technologies (DiSTeBA), University of Salento, 73100 Lecce, Italy; santo.marsigliante@unisalento.it (S.M.); antonella.muscella@unisalento.it (A.M.); 4Laboratory of Cardio-Circulatory, Respiratory, Metabolic and Hormonal Adaptations to Muscular Exercise, Faculty of Medicine Ibn El Jazzar, University of Sousse, Sousse 4000, Tunisia; ezdine_sport@yahoo.fr; 5Institute of Primary Care, University of Zurich, 8006 Zurich, Switzerland; katja@weiss.co.com; 6Medbase St. Gallen Am Vadianplatz, 9001 St. Gallen, Switzerland

**Keywords:** adapted physical activity, locomotors skills, control skills, lability/negativity testing, intervention, swimming

## Abstract

**Simple Summary:**

Swimming can be an alternative in the physical exercise curriculum to improve the motor abilities as well as the social behavior and communication skills of children with autism spectrum disorder (ASD) in schools and institutions. It is important to assess the effects of different forms of aquatic training (e.g., technical vs. game-based) to improve motor and stereotypy skills as well as emotion regulation in autistic children. This information should be of great interest to professionals to choose the appropriate training form to improve each of these abilities. The finding that both forms of swimming had a positive effect on gross motor skills and stereotyped behaviors in autistic children is in agreement with the accumulated evidence of swimming’s effectiveness to alleviate symptoms of motor and behavioral problems. In contrast, emotion regulation is not likely to improve after a short intervention period. The preliminary findings of this study are an important guidance for future researchers to further examine the neurophysiological and cognitive mechanisms of exercise–emotion and exercise–behavior relationships in children with ASD.

**Abstract:**

A variety of aquatic training regimens have been found to be beneficial for individuals with autism spectrum disorders (ASD) in multiple domains. This study investigated and compared the efficacy of two aquatic training regimens (technical vs. game-based) on gross motor skills, stereotypy behavior and emotion regulation in children with ASD. Twenty-two autistic children were randomly assigned into three groups: two experimental groups performed either a technical aquatic program or a game-based aquatic program and a control group. Participants were assessed before and after an 8-week training period, with the Test of Gross Motor Development, the stereotypy subscale of the Gilliam Autism Rating Scale, and the Emotion Regulation Checklist. A significant effect for time was found in gross motor skills and stereotypy behavior in both experimental groups. An improvement in gross motor skills was observed in both experimental groups compared to the control group. A small pre-post change effect in emotion functioning was found in all groups. No significant differences were observed between the experimental groups in all assessed variables. Our findings provide additional evidence suggesting the effectiveness of beneficial effects of aquatic activities on the motor and social skills that underpin the hypothesis that motor and intellectual domains are highly interrelated in autistic children.

## 1. Introduction

Autism spectrum disorder (ASD) has been listed as one of the most common developmental disabilities in the world by the American Psychiatric Association [1]. An increasing number of studies have explored the positive influence of regular physical activity on health-related mental and physical fitness in an individual with developmental and intellectual impairments [2,3,4]. Past studies showed that social behavior is very important in individuals with ASD [5]. Furthermore, motor activities have been researched to promote adaptive functioning and to improve autonomy and participation in social activities among autistic individuals [5,6]. Within the typical symptomatic profile of deficiencies in social communication and interaction across multiple contexts, autistic children showed motor impairments, contributing to the decrease in their socialization abilities [7,8]. In this population, impairment in movement skills was estimated to have a prevalence of 59–79% [9]. From an early age, autistic children showed motor stereotypies, dyspraxia, impairment in early postural control, motor speed and coordination, fluency and balance [10,11,12]. On the other hand, previous research on children and adolescents with autism has associated physical activities (e.g., karate, swimming, basketball) with an improvement in social interaction, communication abilities [13,14,15,16], stereotyped behavior [17], sports skills [16], motor coordination and cardiovascular fitness [18], and quality of life [16,19].

Swimming pool activities have been shown to be an effective way of training psychomotor skills, increasing adaptive behaviors and providing opportunities for social interaction for children with ASD [6,20,21]. In fact, previous studies have demonstrated how structured aquatic activities (i.e., technical modality (TAT) or game-based (GAT)) may improve physical fitness, water orientation, confidence and body awareness, and emotional skills, and provide social interaction opportunities for autistic children [6,20,22,23,24,25,26,27]. In addition, the benefits of the water environment can be explained by the properties of hydrostatic pressure and buoyancy [24,28]. Those properties enable improvements in motor skills as well as in sensory and social behaviors (e.g., paying attention and maintaining eye contact) in autistic children [24,28].

Despite these advantages, few investigations support these conclusions [6,23,28]. Particularly, two studies have shown that a multi-systemic aquatic therapy improves body posture and gestures, personal autonomy, verbal, and non-verbal communication. At the same time, it decreases social, emotional, aggressive, and self-aggressive behavior, and relational and psychomotor deficiencies in autistic individuals [6,23]. 

Within this framework and according to all these findings, it seems important to assess the effect of different forms of aquatic training to prevent benefits of motor and stereotypy skills as well as emotion regulation in autistic children. Such information would be of great interest for professionals in order to choose the appropriate form which could improve each of these abilities.

Given these premises, we hypothesize that different aquatic training regimens can positively affect motor skills and stereotyped behaviors. The purpose of this study was to explore the effects of two 8-week aquatic training programs (TAT vs. GAT) on locomotion, stereotypy skills and emotion regulation in children with ASD.

## 2. Materials and Methods

### 2.1. Study Design

This study adopted a randomized, parallel and controlled pre-to-post measurement design. Children were randomly divided into three groups: two experimental groups performed either a technical aquatic activities program or a game-based aquatic activities program and a control group (CONT). Before starting the experiment, the children were involved in two orientation sessions to familiarize themselves with the experimental procedures. The study was conducted from October 2020 to January 2021 and lasted 8 weeks. The control group was asked to refrain from performing any aquatic activities during the experiment. The participants of the study had no previous experience with the proposed aquatic training. The testing schedule included two sets of tests performed the week before (T1) and the week after the 8-week training period (T2). Participants were assessed with the Test of Gross Motor Development (TGMD-2), the stereotypy subscale of the Gilliam Autism Rating Scale (GARS-2), and the Emotion Regulation Checklist (ERC). All tests were performed on three consecutive days: (1st day) anthropometric measures and TGMD-2; (2nd day) GARS-2; (3rd day) ERC. To avoid any diurnal variation of the participants’ performances, all assessment and training procedures were conducted at the same time of day. Gross motor skills were performed indoors and under similar environmental conditions (temperature: 24 °C). Before the gross motor skills assessment, participants performed a standardized warm-up session, including 10 min of walking and jogging, and 5 min of dynamic stretching. 

All procedures performed in this study were in accordance with the ethical standards of the Declaration of Helsinki [29]. This study was approved by the local research ethics committee (approval No. 015/2020). Parental/legal guardians and participant informed consent were obtained after being thoroughly informed about the purpose, benefits and potential risks of the study. They were also informed that participation was voluntary and that they could withdraw from the study at any time. 

### 2.2. Participants

A sample size power analysis was calculated using the G*Power software (Version 3.1.9.4, University of Kiel, Kiel, Germany) using the F test family (ANOVA: repeated measures, within-between interaction), with 3 conditions (i.e., technical, game-based and control) and 2 times of measurement (T1 and T2). The analysis revealed that 21 subjects would be sufficient to find significant differences (effect size f = 0.40, α = 0.05, statistical power (1-β) = 0.80, correlation of r = 0.5) with an 86.89% (actual power) chance of correctly rejecting the null hypothesis of there being no difference in all assessed variable results across time. 

Inclusion criteria in the present study were: (1) having a certified diagnosis of ASD and meeting the autism diagnostic criteria of the Diagnostic Statistical Manual of Mental Disorders (DSM-V) [1] and having a score > 30 on the Childhood Autism Rating Scale (CARS) [30]; (2) having enough social ability to communicate with instructors and other children; (3) lack of obvious physical or developmental impairments; (4) having no severe behavioral problem; (5) not taking any medications that would interfere with their learning or motor performances; (6) IQ ≥ 80 (measured with verbal subtests of the Wechsler Intelligence Scale for Adolescents) [31]; (7) being able to perform aquatic activities. The exclusion criteria were: (1) having experienced orthopedic injuries or surgeries which limited their movements; (2) lack of parental consent for participation in the study. A total of 62 children with autism were enrolled in a professional autism center in Tunis. A total of 28 of them (21 boys and 7 girls aged 6–7 years) met our inclusion criteria and were recruited (Figure 1).

All participants were from a medium socio-economic level and lived in an urban area with their own families. Children performed two regular physical education lessons per week (1 h per session). They were screened by an experienced physician and were found eligible to participate in our aquatic training programs. Participants were randomly divided into three groups: two experimental groups performed either a technical aquatic activities program (7 boys and 3 girls) or a game-based aquatic activities program (8 boys and 2 girls) and a control group (6 boys and 2 girls, i.e., participated in their regular physical activity). All subjects received the same treatment (i.e., occupational therapy, speech therapy, play therapy) and did not attend any additional exercise training in or out of school during the intervention period.

For 6 participants (2 boys and 4 girls), health issues arose during the experimental period (not caused by the training program), so their data were excluded from further analyses. Therefore, data from 22 participants were used for further analyses (TAT: 7 boys and 1 girl; GAT: 8 boys; CONT: 5 boys and 1 girl). More detailed data about groups is included in Table 1. 

### 2.3. Procedures

#### 2.3.1. Anthropometric Measurements

Height (cm) and body weight (kg) measures were assessed twice for each child, and the mean of each measure set was calculated. Height and body weight measurements were made on a digital scale (OHAUS, Florham Park, NJ, USA) with an accuracy of 0.1 cm and 0.1 kg, respectively. BMI was estimated as the weight divided by the squared height (kg·m^−2^).

#### 2.3.2. Assessment of Gross Motor Skills

The TGMD-2 [32] is an assessment of motor skills in children aged between three and ten and was used to identify children with a significant motor developmental delay. This tool is composed of two subtests aimed at measuring two skill sets: 6 locomotor (LoS) and 6 object control (CoS) skills. Locomotor subset tasks were: (1) sprinting for 15 m; (2) galloping for 10 m; (3) hopping on one leg for 5 m; (4) leaping over an object; (5) performing a horizontal jump; (6) sliding in a straight line. Object control tasks were: (1) striking a stationary ball with a tennis racket; (2) dribbling a basketball; (3) catching a plastic ball; (4) kicking a ball with the preferred foot; (5) throwing a ball with the preferred hand; (6) rolling a ball between 2 cones. Subjects’ performances were videotaped to analyze the movement sequences separately and to assign scores. Children were asked to repeat each task 3 times and were scored as follows: one point was given when the participant performed well twice, whilst a score of 0 was assigned when the participant was unable to perform the test. The sum of raw scores obtained from each subset was used in subsequent analysis. The TGMD-2 showed a good relative reliability (intraclass correlation coefficient = 0.86) [6].

#### 2.3.3. Assessment of Stereotyped Behavior

The stereotypy subscale of GARS-2 [33] was used for the assessment of changes in subjects’ stereotypy severity (Ster). This tool is composed of 14 items and incorporates observations, caregivers’ (parents or teachers) interviews, and questions completed by the examiner according to their interpretation. For each item, caregivers are asked to mark 1 of 4 choices that best express the child’s specific stereotypical behavior using objective assessments based on the frequency of four points (0: the behavior never observed; 1: the behavior seldom observed; 2: the behavior sometimes observed; 3: the behavior frequently observed). The caregivers were asked how often a child: (1) avoids eye contact/looks away; (2) stares at hands or objects; (3) flicks fingers rapidly; (4) eats specific foods; (5) licks, tastes or attempts to eat inedible objects; (6) smells/sniffs objects; (7) whirls or turns in circles; (8) spins objects; (9) rocks back and forth; (10) rapid lunging/darting; (11) prances; (12) flaps hands; (13) makes high-pitched sounds; and (14) slaps, hits, bites self. Caregivers were asked to rate the subject based on the frequency of occurrence of each stereotyped behavior under ordinary circumstances in a 6-h period. Higher scores indicate a higher level of stereotypy. Only the total raw score in the stereotypy subscale of GARS-2 was recorded [17]. The stereotypy subscale is both valid and reliable and has excellent psychometric properties [34].

#### 2.3.4. Assessment of Emotional Regulation

The ERC is a parent report of children’s self-regulation and was designed to evaluate parents’ perspectives on their children’s ability to manage with emotions using a 24-item questionnaire [35]. Items are rated by a parent on a four-point scale assessing the frequency of behaviors (from 1 = never to 4 = always) and yielded two subscales: (1) emotion regulation which assessed expressions of emotions, empathy and emotional self-awareness; and (2) lability/negativity, which assessed mood lability, anger dysregulation and inflexibility. Higher scores on the former subscale indicate more adaptive regulatory processes, whereas higher scores on the latter subscale indicate excessive emotion dysregulation. The ERC is both valid and reliable and has excellent psychometric properties [35].

### 2.4. Training Programs

In the present study, the technical training protocol was based on the Halliwick Method [27] and the foundational swimming skills [26]. The Halliwick Method is divided into four phases: adaptation to water, rotations, movement and control of movement in the water. For basic swimming skills, the target behaviors were flapping kicks, forward crawling strokes and sideways head turns. These skills were selected as being considered essential for learning more sophisticated swimming skills. They also allow the children to be more independent in the water. The game-based training protocol was inspired by the experimental design of Pan [24] and has been detailed in Table 2.

The 8-week training intervention included 16 sessions (2 sessions/week, 50 min/session) at a local indoor swimming pool. Each session included a 5-min general warm-up (e.g., walking, running, jumping jacks, and arm and leg movement), followed by a 7-min warm-up in the pool (e.g., breathing technique, hand and foot movement under the water), 30 min of the selected program and finally, 8 min of cool-down to return to the resting state. The goal of the training routine was to make group sports training competitive and fun. Four qualified swimming trainers in the field of adapted physical education who are certified in working with children with autism were chosen as instructors. Participants practiced in small groups (i.e., 4–8 children) and the instructor/participant ratio of at least 1:2 was maintained. Children were provided with adequate recovery time (i.e., 45–90 s) between exercises and series. Whenever the subject got tired and could not perform an exercise correctly, the exercise was stopped. During the experimental time, the control group participants completed their activities according to their original physical education plans (i.e., different exercises (run, jump, spear, catch, kick and strike), playing with different types of equipment, balance exercises), and their parents were asked not to involve them in any extra motor activity.

### 2.5. Statistical Analysis

Data analyses were performed using SPSS version 20 for Windows and the statistical level was set at 0.05. Values are presented as means ± SD. The normality and the homogeneity for all data before and after the intervention were checked with the Kolmogorov–Smirnov and Levene’s tests, respectively. Compound symmetry was tested using the Mauchly test. The absence of pretest between-group differences suggested a two-way analysis of variance with repeated measures (three-conditions group: (TAT, or GAT or CONT) × time of measurement: (T1 and T2)) was employed to determine the differences between and within the groups. If significant main effects or interactions were present, a Bonferroni post hoc test was conducted. To determine the magnitude of differences, effect sizes (ES) were calculated by converting partial eta squared to Cohen’s d [36]. ES magnitude was classified as trivial (<0.20), small (0.20–0.49), medium (0.50–0.79), and large (0.80 and greater) [36]. Moreover, upper and lower 95% confidence intervals (95% CI) were calculated for the corresponding variation. 

## 3. Results

The normality and the homogeneity of data were confirmed. At baseline, there were no significant differences in demographic, anthropometric, and CARS scores between groups (all *p* > 0.05). Based on contrast, there were no significant intergroup differences in gross motor skills, stereotypy behavior and emotion regulation raw scores in the pretest (all *p* > 0.05). The absolute values resulting from the analyzes between groups and within the same group are shown in Table 3.

### 3.1. Gross Motor Skills

Observing the locomotor’s ability, subjects increased their proficiency, from baseline to post-test, in a different way (F_1,19_ = 127.959; *p* < 0.0001, ES = 2.459). A significant interaction was observed between the conditions and time (F_2,19_ = 25.995; *p* < 0.0001, ES = 1.507) in which raw scores improved from pre- to post-test for both experimental conditions (Table 3), and both experimental conditions showed higher post-test raw scores than CONT (95% CI = 3.984 to 10.016 and 1.484 to 7.516; ES = 3.834 and 2.047; all *p* < 0.005; for TAT and GAT, respectively).

Moreover, regarding the control skills, there was a main effect for time (F_1,19_ = 284.548; *p* < 0.0001, ES = 3.674) in which raw scores improved from pre- to post-test (*p* < 0.0001). A significant interaction was observed between condition and time (F_2,19_ = 44.360; *p* < 0.0001, ES = 1.985) in which raw scores improved from pre- to post-test for both experimental conditions (Table 3), and both experimental conditions showed higher post-test raw scores than CONT (95% CI = 2.286 to 10.380 and 2.036 to 10.130; ES = 2.375 and 1.950; all *p* < 0.005; for TAT and GAT, respectively).

### 3.2. Emotion Regulation

For the emotion regulation, there was no statistical interaction between condition and time (F_2,19_ = 0.826; *p* = 0.453, ES = 0). However, a main effect for time (F_1,19_ = 26.272; *p* < 0.0001, ES = 1.097) was observed from pre- to post-test across all conditions (*p* < 0.0001) (Table 3).

For lability and negativity, there was a main effect for time (F_1,19_ = 10.120; *p* = 0.005, ES = 0.659) in which raw scores improved from pre- to post-test (*p* = 0.005) (Table 3). However, there was no statistical interaction between condition and time (F_2,19_ = 1.641; *p* = 0.220, ES = 0.241).

### 3.3. Stereotypy Behavior

For stereotyped behavior, a significant interaction was observed between condition and time (F_2,19_ = 5.347; *p* = 0.014, ES = 0.629). Moreover, a large magnitude and statistically significant main effect for time was observed (F_1,19_ = 200.479; *p* < 0.0001, ES = 3.082) in which scores decreased from pre- to post-test across all groups (*p* < 0.0001) (Table 3).

## 4. Discussion

Previous studies on the effects of physical activity in people with ASD have shown that learning motor skills is useful for enhancing motor and functional skills [20,37] and have also suggested that they could be useful for improving behavior, and surmounting cognitive and social difficulties [18,38,39,40]. Regarding aquatic exercise interventions, several results have shown an improvement in motor skills and physical fitness [4], while only a few empirical pieces of evidence have suggested the usefulness of this strategy in the treatment of functional disabilities of individuals with autism [24,27,41,42].

This investigation aimed to examine the efficacy of two aquatic-based interventions (TAT vs. GAT) compared to an active control intervention for improving gross motor skills, stereotyped behaviors, and emotion regulation in autistic children. Results showed that both aquatic training interventions were effective in improving locomotors and control skills in subjects with ASD. However, smaller changes in emotional functioning, with respect to the control group, were observed. These improvements are coherent with prior studies that showed that drills in water promote various aspects of gross motor proficiency and coordination skills in individuals with ASD [6,23,24,25,27].

Regarding locomotion, Yilmaz et al. [27] reported that 10 weeks of an aquatic training program was successful in improving running and hopping on one leg, horizontal jumps, and sliding. Aquatic-based interventions also improve physical fitness (i.e., balance, speed, agility, power, muscle strength, flexibility, coordination, cardiorespiratory endurance), aquatic orientation and object control skills (catch a ball with a tennis racket, stationary bounce, catch, kick, and overhand throw), in children with ASD [27]. To perform complex motor patterns (object control patterns), the subject is asked to process afferent information quickly and efficiently [43,44]. Indeed, Battaglia et al. [6] attributed the increased control skills to a possible interaction between perceptual capacity, visual-motor integration, and motor skills; exercises in water may be eligible to stimulate these skills.

Although the children in the control group practiced other sports (but not swimming), the results showed no improvement in gross motor skills. This data could confirm that swimming can induce complex adaptations of the nervous system and favor the transmission of neural impulses [45]. Since water density is 800 times greater than air density, exercise in water may increase the muscle strength through a resistive medium without extreme weight-bearing joints [6].

Furthermore, in our study, the training programs for both experimental groups were designed with care and considered the different variables for each child (i.e., age, disability, water experience and play skills). In addition, during the experimentation, the targets were revised with the addition of dynamic tasks that were more challenging, including rotation, balance and controlled movement, and independent movement in the water. Hence, these well-prepared interventions are more suitable for children with ADS than for other ordinary training programs without specific purposes.

Previous studies had reported that different training programs (including swimming) could be effective and stimulate processes improving stereotypical behavior [17,27]. Here, we had shown in T2 a significant reduction in the frequency of stereotypy in all groups. Indeed, it was suggested that the comparison of physical stimulus obtained through physical activities to that obtained through stereotyping for children with autism can lead to stereotyping reductions [39]. Socialization was a crucial element of our programs for children with ASD. Likewise, it was demonstrated that the reduction in stereotypical behavior is in relation to physical activities [46,47], which also contributes to social integration in children with ASD, through its effect on reducing behavior problems [46]. Good interaction between each child and coach was noted with the eye contact, synergy during games or technical exercises and paying attention.

Therefore, the progress in the control group and in the experimental groups, in stereotyped and emotional behavior, demonstrates the greater importance of the social aspect widely implemented in training programs, compared to the type of training itself. Only two studies reported emotional changes in children with ASD after aquatic therapy intervention; implementation of a 10-week water program on children with ASD resulted in larger positive changes in social and physical functioning, and smaller changes in emotional and school functioning [23,42]. Therefore, our results, which show that in T2 there are no evident differences in the emotional response in the experimental groups compared to the controls, would, at the moment, be consistent with the previous observations.

In conclusion, our results showed that the aquatic training programs, regardless of the methodologies, were effective in children with ASD. In fact, children with movement difficulties experience more success in attaining movement skills and control skills in an aquatic environment compared to a land-based setting. The buoyancy of water and its decreased gravity effects may allow an individual to exercise motor skills more easily. In addition, pools, especially those in community settings, offer an important occasion for participation and social interaction [6,25].

We recognize an important limitation of this study. Although the statistical analysis revealed that a total sample size of 21 would be sufficient to find significant differences, objectively, the number of participants is still relatively small. However, the results encourage future randomized trials with larger sample sizes carefully controlled also for sex differences and for other relevant demographic variables. Finally, the survey lasted only 8 weeks, and we recognize that longer training periods may be required to achieve greater emotion regulation and stereotypy skills.

Our results showed the effectiveness of both training programs in enhancing several functional behaviors as well as swimming skills of children with ASD, suggesting that aquatic-training treatment is a viable approach for promoting positive changes in relevant aspects of ASD. Therefore, aquatic therapies can be effectively combined with the standard-of-care treatment of ASD, not only for its physical but also for its social, emotional, and functional effects.

## 5. Conclusions

Overall, the findings provide additional evidence suggesting the effectiveness of the beneficial effects of aquatic activities on motor and social skills, supporting the hypothesis that the motor and intellectual domains are highly interrelated in individuals with ASD. Although the present study does not provide definitive proof of the efficacy of aquatic training in treating emotional impairments in autism, it suggests that both training modalities are viable approaches for enhancing gross motor skills in autistic children. Our results suggest that both training programs could be implemented in the educational and vocational treatment of children with ASD to promote not only their physical but also their social, emotional, and functional effects.

## Figures and Tables

**Figure 1 biology-11-00657-f001:**
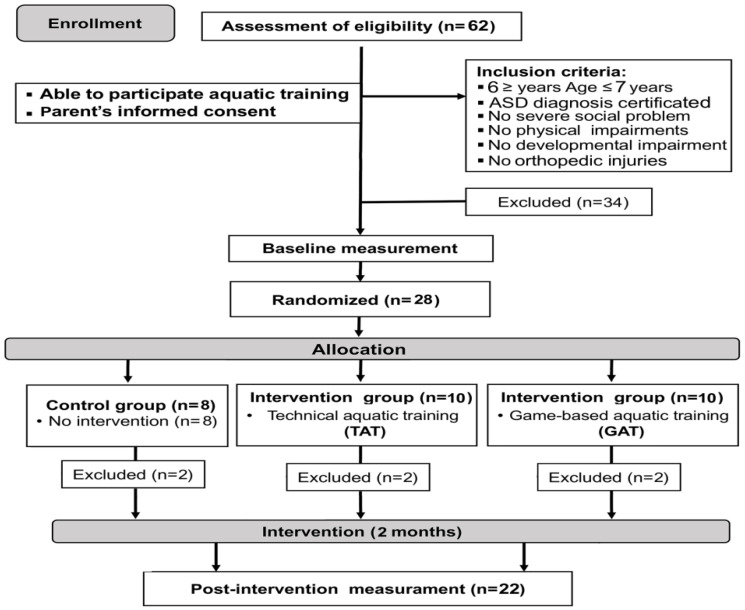
CONSORT diagram of participants’ recruitment, allocation, follow-up and analysis.

**Table 1 biology-11-00657-t001:** Demographic/anthropometric characteristics and CARS scores for the experimental (technical aquatic training, *n* = 8; game-based aquatic training, *n* = 8), and control (*n* = 6) groups.

	TAT (7 B + 1 G)	GAT (8 B)	CONT (5 B + 1 G)
Age (year)	6.3 ± 0.5	6.4 ± 0.5	6.3 ± 0.5
Height (cm)	113.1 ± 11.3	113.2 ± 3.3	112.8 ± 2.4
Weight (kg)	18.7 ± 3.5	18.5 ± 2.2	18.3 ± 3.2
BMI (kg·m^−2^)	14.7 ± 2.8	14.4 ± 1.7	14.4 ± 2.4
IQ	83.9 ± 2.1	83.4 ± 2.6	84.3 ± 1.4
CARS	35.1 ± 2.0	35.9 ± 2.7	34.8 ± 2.6

Values are given as mean ± SD; TAT: technical aquatic activities program group; GAT: game-based aquatic activities program group; CONT: control group; B: boys; G: girls; BMI: body mass index; IQ: intelligence quotient; CARS: Childhood Autism Rating Scale.

**Table 2 biology-11-00657-t002:** Description of the 8-week game-based training protocol.

Week	Session	Sets/Repetitions	Aim	Activities/Games
1st	1	6 × 2	- Social interaction - Aquatic and motor/control skills development - Communication - Cooperation - Water confidence	Cooperative games/activities (e.g., noodle kick/jump/float, Aqua hot potato, Dinosaur eggs)
	2	6 × 2		
2nd	3	6 × 2		
	4	6 × 2		
3rd	5	7 × 2		Cooperative games/activities (e.g., hula-hoop swimming, throwing, and catching the ball, a star fish is born)
	6	7 × 2		
4th	7	7 × 2		
	8	7 × 2		
5th	9	8 × 2		Cooperative and fun games (e.g., circle tag, ball hoop and block, boogie-woogie)
	10	8 × 2		
6th	11	8 × 2		
	12	8 × 2		
7th	13	6 × 2		Cooperative and fun games (e.g., Octopus tag, water polo)
	14	6 × 2		
8th	15	6 × 2		
	16	6 × 2		

**Table 3 biology-11-00657-t003:** Baseline (T1) and final (T2) measures of locomotor and control skills, emotion regulation and stereotypy subscale scores for the experimental (technical aquatic training, *n* = 8; game-based aquatic training, *n* = 8), and control (*n* = 6) groups.

Variables	Group	T1	T2	ES	CI 95%
LoS scores	TAT	6.75 ± 1.75	11.0 ± 2.0 ^†,‡^	2.260	3.35–5.15
	GAT	4.0 ± 2.73	8.5 ± 2.56 ^†,‡^	2.928	3.60–5.40
	CONT	3.8 ± 2.64	4.0 ± 1.55	-	-
CoS scores	TAT	7.0 ± 3.66	13.0 ± 2.33 ^†,‡^	1.952	4.94–7.06
	GAT	4.38 ± 3.42	12.75 ± 3.15 ^†,‡^	2.547	7.32–9.43
	CONT	5.5 ± 3.0	6.7 ± 3.1	-	-
EmR scores	TAT	27.5 ± 3.07	28.13 ± 3.0 ^†^	0.206	0.05–1.20
	GAT	27.5 ± 3.07	28.63 ± 2.92 ^†^	0.385	0.55–1.70
	CONT	26.33 ± 2.42	27.17 ± 2.4 ^†^	0.346	0.17–1.50
L/N scores	TAT	25.5 ± 3.17	24.0 ± 3.51 ^†^	0.455	0.45–2.55
	GAT	24.63 ± 3.29	23.38 ± 3.29 ^†^	0.380	0.20–2.30
	CONT	24.5 ± 3.45	24.33 ± 3.01	-	-
Ster scores	TAT	31.63 ± 5.32	28.25 ± 5.37 ^†^	0.632	2.60–4.15
	GAT	34.63 ± 4.53	30.63 ± 4.47 ^†^	0.889	3.23–4.77
	CONT	32.17 ± 3.87	30.0 ± 3.69 ^†^	0.573	1.27–3.06

Values are given as mean ± SD; LoS scores: locomotors skills raw scores; CoS scores: control skills raw scores; EmR scores: emotion regulation scores; L/N: lability/negativity scores; Ster scores: stereotypy subscale scores; TAT: technical aquatic activities program group; GAT: game-based aquatic activities program group; CONT: control group; ES: effect size; CI 95%: 95% confidence intervals. ^†^ A significant difference when comparing T1 and T2. ^‡^ Significantly different from CONT at T2. The statistical level was set at *p* ≤ 0.05.

## Data Availability

The data presented in this study are available on request from the corresponding author. The data are not publicly available due to privacy reasons.

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
