# Peer review of "Effects of Aquatic Training in Children with Autism Spectrum Disorder"

_biology, 2022, doi:10.3390/biology11050657_

Round 1

Reviewer 1 Report

I read a very interesting article entitled "Effects of Aquatic Training in Children with Autism Spectrum Disorder". It is a very original study with good research design and good methodology. The article may be published after minor corrections. 
-In the names of the authors (line 4) what does (which refers to) the symbol  † ?
-Because the number of children who participated in the research is small, the authors should give more information about the variables they introduced in G * Power.
-Line 143:  “For 7 participants (2 boys and 4 girls),” please correct.
- In Table 1 I would like the authors to add the gender of the participants.

Author Response

Referee 1

Main comments:

I read a very interesting article entitled "Effects of Aquatic Training in Children with Autism Spectrum Disorder". It is a very original study with good research design and good methodology. The article may be published after minor corrections. 

Author's response:

We thank the referee for his comments.

Comment 1

In the names of the authors (line 4) what does (which refers to) the symbol  † ?

Author's response:

The symbol ''†'' was deleted. Please find changes in the text.

Comment 2

Because the number of children who participated in the research is small, the authors should give more information about the variables they introduced in G * Power.

Author's response:

Thank you for your valuable comment. Sample size power was rewritten and all details were added as follow:

'' A sample size power analysis was calculated using the G*Power software (Version 3.1.9.4, University of Kiel, Kiel, Germany) using the F test family (ANOVA: repeated measures, within-between interaction), with 3 conditions [i.e., technical (TAT), game-based (GAT) and control (CONT)] and 2 times of measurement (T1 and T2). The analysis revealed that 21 subjects would be sufficient to find significant differences (effect size f = 0.40, α = 0.05, statistical power (1-β) = 0.80, correlation of r = 0.5) with a 86.89% (actual power) chance of correctly rejecting the null hypothesis of there being no difference in all assess variable results across time.

Please find changes in the text.

Comment 3

- Line 143:  “For 7 participants (2 boys and 4 girls),” please correct.

Author's response:

We thank very much the expert referee for his comment.

The number was corrected. Please find changes in the text.

Comment 4

- In Table 1 I would like the authors to add the gender of the participants.

Author's response:

We thank very much the expert referee for his comment.  All recommendations were added. Please find changes in the text.

Reviewer 2 Report

First of all, I would like to thank the authors and the journal for giving me the opportunity to review this work, which I found extremely interesting. When we think about the difficulties that children with neurodevelopmental disorders have, we may not give sufficient importance to aspects related to motor coordination, which are essential for the autonomy of the individual.In this work the Dependent Variables are well designed, taking into account the importance of gross motor skills, stereotypies and emotional self-regulation.In this work the Dependent Variables are well designed, taking into account the importance of gross motor skills, stereotypies and emotional self-regulation. The following aspects are only intended to improve the quality of the paper and contribute to make it more understandable for future readers.

The number of participants, 28 in the pretest and only 22 in the posttest including the control group, is a very small sample, as the authors themselves indicate as one of the limitations of their study. It is very difficult to recruit this type of sample of boys/girls with ASD and, in addition, the authors have been very careful when selecting and describing their participants, who are relatively homogeneous in their anthropometric indicators and in variables such as age or socioeconomic level.

However, when processing the data, they seem to have forgotten about the size of the subgroups and resorted to parametric statistics. It is true that they have assessed some preconditions of the data, such as the adjustment to normality and their random nature, but it is at least questionable to use ANOVA to establish intragroup differences when the sizes of the groups are 8, 8 and 6 participants in the post-test measurement. I understand that this decision is inappropriate and, authors should justify it in more detail.

On the other hand, when they speak of differences between the experimental groups and the control group, they do so without offering evidence to indicate that these differences did not already exist in the pretest. Using an ANCOVA you can use the pretest score as a covariate to adjust for baseline values in analyzing a posttest scores or, directly, showing the absence of significant intergroup differences in the pretest based on a contrast. This aspect deserves the most attention from a methodological point of view.

On the other hand, in different sections of the article, including the abstract and conclusions, the authors seem to refer to the Stereotypies of individuals with ASD and their social skills as if they were interchangeable variables. This should be corrected or adequately justified. In addition to these major errors, I have detected minor flaws that the authors should review, such as :

In the introduction section on line 58 one can read "motor diseases were estimated to have a prevalence from 59-79%". When the article taken as a reference (Green et al. 2009) refers to "Impairment in movement skills", the term "motor diseases" could lead to confusion. Especially if we take into account that the authors consider as inclusion criteria in the study "lack of obvious physical or developmental impairments (line 122) and as exclusion criteria (line 126) "having experienced orthopedic injuries or surgeries which limited their movements". Perhaps the authors should be more explicit in all these terms used in relation to motor skills.

Finally, certain errors were detected that suggest the need to revise the text once again, one of them would be the one that appears in line 143 where it says "7 participants (2 girls and 4 boys)".

Author Response

Referee 2

Comment 1

The number of participants, 28 in the pretest and only 22 in the posttest including the control group, is a very small sample, as the authors themselves indicate as one of the limitations of their study. It is very difficult to recruit this type of sample of boys/girls with ASD and, in addition, the authors have been very careful when selecting and describing their participants, who are relatively homogeneous in their anthropometric indicators and in variables such as age or socioeconomic level.

However, when processing the data, they seem to have forgotten about the size of the subgroups and resorted to parametric statistics. It is true that they have assessed some preconditions of the data, such as the adjustment to normality and their random nature, but it is at least questionable to use ANOVA to establish intragroup differences when the sizes of the groups are 8, 8 and 6 participants in the post-test measurement. I understand that this decision is inappropriate and, authors should justify it in more detail.

Author's response:

We thank very much the expert referee for his comment.

As recommended by Hopkins et al. (2009): avoid purely nonparametric analyses. A requirement for deriving inferential statistics with the family of general linear models is normality of the sampling distribution of the outcome statistic (condition assumed in all variables at pre-test). Although there is no test that data meet this requirement, the central-limit theorem ensures that the sampling distribution is close enough to normal for accurate inferences, even when sample sizes are small (~10) and especially after a transformation that reduces any marked skewness in the dependent variable or nonuniformity of error. Testing for normality of the dependent variable and any related decision to use purely nonparametric analyses (which are based on rank transformation and do not use linear or other parametric models) are therefore misguided. Such analyses lack power for small sample sizes, do not permit adjustment for covariates, and do not permit inferences about magnitude.

HOPKINS, WILLIAM G., MARSHALL, STEPHEN W., BATTERHAM, ALAN M., HANIN, JURI. Progressive Statistics for Studies in Sports Medicine and Exercise Science, Medicine & Science in Sports & Exercise: January 2009 - Volume 41 - Issue 1 - p 3-12

Comment 2

On the other hand, when they speak of differences between the experimental groups and the control group, they do so without offering evidence to indicate that these differences did not already exist in the pretest. Using an ANCOVA you can use the pretest score as a covariate to adjust for baseline values in analyzing a posttest scores or, directly, showing the absence of significant intergroup differences in the pretest based on a contrast. This aspect deserves the most attention from a methodological point of view.

Author's response:

Thank you for your valuable comment.  ''Based on a contrast, there were no significant intergroup differences in gross motor skills, stereotypy behavior, and emotion regulation raw scores in the pretest (all p > 0.05).'' This sentence was added in the result section. Please find changes in the text.

Absence of pretest between-group differences suggested 2-way analysis of variance with repeated measures [3-conditions group: (TAT, or GAT, or CONT) × time of measurement: (T1 and T2)] was employed to determine the differences between and within-groups. The red part of the sentence was added in the statistical analysis section. Please find changes in the text.

Comment 3

On the other hand, in different sections of the article, including the abstract and conclusions, the authors seem to refer to the Stereotypies of individuals with ASD and their social skills as if they were interchangeable variables. This should be corrected or adequately justified.

Author's response:

Unfortunately, we don't understand the expert referee's request; we checked sentences where we indicated both stereotypies and social skills as interchangeable variables, but we didn't find them. However, although not interchangeable variables, stereotypy (specially high levels of stereotypy) in individuals diagnosed with ASD were correlated with more significant impairments in social. Reducing stereotypy may thus potentially occasion an increase in appropriate social and adaptive behaviors.

Comment 4

In the introduction section on line 58 one can read "motor diseases were estimated to have a prevalence from 59-79%". When the article taken as a reference (Green et al. 2009) refers to "Impairment in movement skills", the term "motor diseases" could lead to confusion. Especially if we take into account that the authors consider as inclusion criteria in the study "lack of obvious physical or developmental impairments (line 122) and as exclusion criteria (line 126) "having experienced orthopedic injuries or surgeries which limited their movements". Perhaps the authors should be more explicit in all these terms used in relation to motor skills.

Author's response:

We thank very much the expert reviewer for his comment. The term ''motor diseases'' was changed by ''impairment in movement skills''. Please find changes in the text.

Comment 5

Finally, certain errors were detected that suggest the need to revise the text once again, one of them would be the one that appears in line 143 where it says "7 participants (2 girls and 4 boys)".

Author's response:

We thank very much the expert reviewer for his comment.

The number was corrected. Please find changes in the text.

Reviewer 3 Report

Review Report:

Journal: Biology

Manuscript ID: biology-1651004

Title:

Effects of aquatic training in children with Autism Spectrum Disorder.

Rating the manuscript:

Overall Recommendation: Accept after minor Revisions: The paper is in principle accepted after revision based on the reviewer’s comments.

Review Report:

In this review, the authors compared two technical vs. game-based acquatic training on gross motor skills, stereotype behavior, and emotion regulation in children with ASD. In their study, 22 autistic children were randomly assigned into three groups: two experimental groups performed either a technical aquatic program or game-based aquatic program and a control group. The authors evaluated that the participants were assessed before and after an 8-week training period, with the test of gross motor development, the stereotypy subscale of Gilliam Autism Rating Scale, and the emotion regulation checklist. A significant effect for time was found in gross motor skills and stereotypy behaviour in both experimental groups. An improvement of gross motor skills was observed in both experimental groups compared to the control group. A small pre-post change effect in emotion functioning was found in all groups. No significant differences were observed between the experimental groups in all assessed variables. Their findings provide additional evidence suggesting the effectiveness of beneficial effects of aquatic activities on the motor and social skills that underpin the hypothesis that motor and intellectual domains are highly interrelated in autistic children.

However, there are few points that need to be addressed in the paper:

  1. In Figure 1, and all tables, please add few lines to describe the figures and tables.
  2. In the abstract and the introduction, they can describe in one paragraph, the essence of this study, more in terms of clinical application. This will add more value to this study.
  3. Table 2 fonts need to be increased.

Author Response

Referee 3

Main comments:

Accept after minor Revisions: The paper is in principle accepted after revision based on the reviewer’s comments.

Comment 1

In Figure 1, and all tables, please add few lines to describe the figures and tables.

Author's response:

We thank the referee for his comment. All recommendations were added.

Please find changes in the text.

Comment 2

In the abstract and the introduction, they can describe in one paragraph, the essence of this study, more in terms of clinical application. This will add more value to this study.

Author's response:

Thank you for your valuable comment. The following paragraph was added in the introduction: " Within this framework and according to all these findings, it seems important to assess the effect of different forms of aquatic training to prevent benefits of motor and stereotypy skills as well as emotion regulation in autistic children. Such information would be of great interest for professionals in order to choose the appropriate form which could improve each of these abilities.''

Please find changes in the text.

Comment 3

Table 2 fonts need to be increased.

Author's response:

We thank very much the expert referee for his comment. Fonts was increased.

Please find changes in the text.

Round 2

Reviewer 2 Report

Dear authors,

As I told you in my first review, I find your study interesting and relevant and it shows that you have paid attention to some relevant details in the design of an experimental study. I have found that you have read my assessments and have partially taken some of them into account. But not most of them and, above all, they have not paid attention to the most relevant ones.

I am sorry to be insistent in my remarks about the treatment of the data. But as you understand, and as you have included among the limitations of your study, the number of participants you have used is very small. Regardless of whether this small sample size is justified by the difficulty of accessing special samples, the data you have obtained present serious limitations to your study and require specific statistical treatment. I would like to point out some errors that occur in your study and make it unpublishable, in its current presentation.

When we have few data, normality tests lose statistical power and we cannot be sure of the type of distribution of the data. However, for non-parametric tests the sample size should not be too small either. Normality tests should be done on a group-by-group basis (with 8, 8 and 6 participants) not adding them all up to n=22. By breaking down into such small groups, the claim of "normality" of the distributions becomes meaningless.

On the other hand, they have used the Kolmogorov-smirnov test to establish the normality of the distribution, when the correct test would have been the Shapiro-Wilk test.

To justify their use of parametric tests in this case, they have resorted to citing one author who recommends it. However, that article also recommends the use of large samples:

Whatever approach you use, sample size needs to be quadrupled to adequately estimate individual differences or responses and effects of covariates on the main effect. Larger samples are also needed to keep clinical error rates for clinical or practical decisions acceptable when there is more than one important effect in a study (Note 3). See Hopkins (12) for a spreadsheet and details of these and many other sample-size issues.

I encourage you to expand the number of participants in the study and to reattempt publication in this or a similar journal. (Hopkins et al., 2009).